# Megakaryocyte- and Platelet-Derived Microparticles as Novel Diagnostic and Prognostic Biomarkers for Immune Thrombocytopenia

**DOI:** 10.3390/jcm11226776

**Published:** 2022-11-16

**Authors:** Wen Wang, Bin Zuo, Yao Wang, Xinyu Li, Zhen Weng, Juping Zhai, Qingyu Wu, Yang He

**Affiliations:** 1NHC Key Laboratory of Thrombosis and Hemostasis, Jiangsu Institute of Hematology, The First Affiliated Hospital of Soochow University, Suzhou 215006, China; 2MOE Engineering Center of Hematological Disease, Cyrus Tang Hematology Center, Collaborative Innovation Center of Hematology, Soochow University, Suzhou 215123, China; 3Department of Blood Transfusion, The First Affiliated Hospital of Soochow University, Suzhou 215006, China

**Keywords:** immune thrombocytopenia, platelet microparticles, megakaryocyte microparticles, biomarker, diagnosis

## Abstract

Altered cell-derived microparticles (MPs) have been reported in multiple autoimmune diseases. However, the roles of megakaryocyte- and platelet-derived MPs (MKMPs and PMPs) in immune thrombocytopenia (ITP) have not been investigated. In this study, we examined plasma MKMP and PMP levels in patients with ITP and evaluated their potential diagnostic values. Plasma MKMP and PMP levels were analyzed by flow cytometry in a discovery set of ITP patients (*n* = 78), non-immune thrombocytopenia (TP) patients (*n* = 69), and age- and gender-matched healthy controls (*n* = 88). Samples from a therapy set of ITP patients (*n* = 21) were used to assess the response to thrombopoietin receptor agonist (TPO-RA) treatment. Spearman correlation analysis was performed between MP levels and disease parameters. Receiver operator characteristic (ROC) curves were generated to evaluate the diagnostic values of the MPs. We found that plasma MKMP and PMP levels were significantly lower in ITP patients than those in healthy controls (*p* values < 0.0001) but higher than in those in TP patients (*p* < 0.002 and *p* < 0.0002, respectively). After normalization to platelet counts, PMP/Platelet ratios in ITP patients were higher than those in TP patients and healthy controls (*p* values < 0.001). PMP/Platelet ratios had a diagnostic value for ITP (area under the curve = 0.808, *p* < 0.0001) with 73.1% sensitivity and 77.3% specificity. MKMP levels can be used to discriminate ITP from TP with a cut-off value of 112.5 MPs/μL and a sensitivity of 74.4%. Moreover, both MKMP and PMP levels were elevated in ITP patients who responded to TPO-RA treatment. Plasma PMP levels positively correlated with platelet counts in the responders (*r* = 0.558, *p* < 0.01). Our results indicate that plasma MKMP and PMP levels are decreased in ITP patients and that plasma MKMP and PMP levels may serve as biomarkers for ITP diagnosis and prediction of TPO-RA treatment response.

## 1. Introduction

Immune thrombocytopenia (ITP) is an autoimmune disease characterized by increased platelet destruction and impaired platelet production. The main clinical manifestation of ITP includes isolated thrombocytopenia with platelet counts < 100 × 10^9^/L and the absence of other cytopenias [1]. The disease is associated with an increased risk of bleeding. In recent years, advances in the understanding of disease pathogenesis have greatly improved the management of ITP. However, a significant proportion of ITP patients remain refractory with bleeding that could be life-threatening [2,3]. Currently, there is no “gold standard” or specific biomarkers for ITP diagnosis and assessment of treatment response [4].

Circulating microparticles (MPs) are 0.1–1 μm extracellular vesicles (EVs) shed from cells in response to activation or stress. Depending on their cellular origins, MPs carry specific sets of proteins, lipids, and RNAs, which may mediate intercellular communications [5]. Platelet-derived microparticles (PMPs) and megakaryocyte-derived microparticles (MKMPs) share some common surface markers, such as the glycoproteins CD41 and CD42b and phosphatidylserine; however, they also can be distinguished by CD62P expression on PMPs [6]. Studies have shown that PMPs contribute to blood coagulation, inflammation, and arterial thrombosis and may serve as potential differentiation biomarkers in cancers [7,8,9]. Engineered PD-1-expressing PMPs have been used in cancer immunotherapy [10]. Meanwhile, elevated PMP levels have been reported in ITP and confer procoagulant properties in some ITP patients [11,12,13]. MKMPs, released during the megakaryocyte maturation [6], are found to induce platelet production without thrombopoietin (TPO) supplementation [14,15]. These findings suggest that MKMPs may serve as biomarkers for tracking megakaryocyte dynamics and potential treatment for ITP. Previous bone marrow smears and ultrastructural analyses have also indicated compromised megakaryocyte maturation in ITP patients [16]. To date, the extent to which MKMPs are altered in ITP remains poorly understood.

In this study, we hypothesized that MKMP and PMP levels may be altered in ITP patients and that the altered MKMP and PMP levels may serve as diagnostic and prognostic biomarkers for ITP. We employed flow cytometry to quantify plasma MKMP and PMP levels in healthy controls and patients with ITP and non-immune thrombocytopenia (TP). Our results indicated that plasma MKMP and PMP levels were decreased in ITP and TP patients compared to those in healthy controls and that the levels in ITP patients were higher than those in TP patients. Moreover, our results suggest that plasma MKMPs and PMPs may serve as biomarkers for ITP diagnosis and the response to TPO receptor agonist (TPO-RA) treatment.

## 2. Materials and Methods

### 2.1. Study Design and Subjects

This case-control study was conducted in the Hematology Department of The First Affiliated Hospital of Soochow University between June 2021 and January 2022. A total of 78 ITP and 69 TP patients and 88 age- and gender-matched healthy controls were included as a “discovery” set to determine the differences in MP levels and evaluate their potential value for ITP diagnosis. The platelet counts in TP and ITP patients were <100 × 10^9^/L. The diagnosis of ITP was based on the guideline of the American Society of Hematology [17]. Based on disease duration, ITP patients were divided into three subgroups, i.e., newly-diagnosed (<3 months of diagnosis), persistent (3–12 months), and chronic (duration ≥ 12 months). According to the results from the flow cytometric immunobead assay (FCIA) [18,19], ITP patients were also divided into antibody-positive and negative subgroups. Findings from the discovery set were further analyzed in another “therapy” set of additional ITP patients (*n* = 21) to evaluate MP level changes in response to a two-week TPO-RA treatment (eltrombopag or avatrombopag). Responders were defined as platelet counts ≥ 30 × 10^9^/L or doubling of the baseline platelet counts and absence of bleeding 2 weeks after the treatment. Non-responders were defined as platelet counts < 30 × 10^9^/L or <2-fold increase in baseline platelet counts and with bleeding. None of the ITP patients received any drug treatment one month prior to this study. Among the 69 TP patients, 38 had acute leukemia, 15 had myelodysplastic syndrome, 9 had secondary thrombocytopenia, and 7 had pancytopenia. The study was approved by the institutional review board of the First Affiliated Hospital of Soochow University (approval #2020-452). Written informed consent was obtained from all participants and the study was conducted in accordance with the Declaration of Helsinki.

### 2.2. Plasma Collection and Preparation

Plasma samples were collected according to the International Society for Extracellular Vesicles (ISEV) recommendations [20]. Briefly, peripheral venous blood was collected in a 2-mL EDTA-anticoagulant tube using a 21-gauge needle. The first 2 mL of blood was discarded to avoid platelet activation during venipuncture. The tubes were kept vertically at room temperature for <2 h [21]. To minimize in vitro MP generation, platelet-free plasma (PFP) was obtained by centrifugation (2500× *g*, 15 min) to eliminate blood cells. PFP was then centrifuged (2500× *g*, 15 min) to pellet residual platelets and cellular fragments [22]. MPs in PFP were analyzed immediately.

### 2.3. Quantification of MPs by Flow Cytometry

Latex beads (0.5 μm and 1.0 μm) (Sigma Aldrich, L5530 and L9654, St. Louis, MO, USA) were used to characterize the MP gate by Gallios flow cytometer with the W^2^FSC mode (Beckman Coulter, Brea, CA, USA) (Figure 1A, top panels).

Processed PFP (40 μL) was diluted with 60 μL of 0.1 μm-filtered Annexin V binding buffer (BD Biosciences, 556454) and incubated with FITC-Annexin V, PE-CD41 antibody, and APC-CD62P antibody (BD Biosciences, 556419, 558040, and 550888, Franklin Lakes, NJ, USA) at room temperature for 1 h in the dark and then diluted to 500 μL with the buffer [22]. Counting beads (4 μm) (Invitrogen, C36995, Waltham, MA, USA) were added before sample loading onto flow cytometry. Calcium-free phosphate buffer was used as a negative control for Annexin V labeling. To ensure immunostaining specificity for MPs and reduce the background, PFP supernatant and pellets obtained after 1 h ultracentrifugation at 21,000× *g*, 4 °C, as well as 2% Triton X-100 treated PFP were used as gating controls (Appendix A). MP events were recorded under the fluorescence parameter 1 (FL1) discrimination setting, which detected more MP events than under the forward light scatter (FS) discrimination setting [23,24] (Figure 1B). Fluorescent channels were set at a logarithmic gain. Data were acquired for 5 min per sample. MPs were defined as events with size < 1 μm and binding to Annexin V. PMPs and MKMPs were defined as CD41^+^ CD62P^+^ and CD41^+^CD62P^−^ MPs, respectively (Figure 1A, bottom panels) [6]. MP levels were calculated by counting beads (1000 per μL) using the following formula. PMPs (MPs/μL) = Annexin V^+^ CD41^+^CD62P^+^ number × (beads added/beads counted)/40 μL; MKMPs (MPs/μL) = Annexin V^+^ CD41^+^CD62P^−^ number × (beads added/beads counted)/40 μL. Data were analyzed using Kaluza software (version 1.2, Beckman Coulter, Brea, CA, USA).

### 2.4. Statistical Analysis

Continuous variables were described as median values (interquartile range) and categorical variables were presented as numbers (percentage). The Mann–Whitney U test and one-way ANOVA test were used for two- and multiple-group comparisons, respectively. Changes in plasma MP levels before and after treatment were compared using the paired-samples Wilcoxon test. The Spearman correlation analysis was used to determine the correlation between plasma MP levels and other blood parameters. Receiver-operator characteristic (ROC) curves and the area under the curve (AUC) values were used to assess the sensitivity and specificity of MPs in disease diagnosis. All statistical analyses were performed using GraphPad Prism 8 (GraphPad Software, San Diego, CA, USA). *p* < 0.05 was considered statistically significant.

## 3. Results

### 3.1. Study Population

The characteristics of all participants were shown in Table 1. There were no differences in the age and gender distribution of the participants. Platelet counts in ITP and TP patients were significantly lower than those in healthy controls (both *p* values < 0.0001). Samples of a therapy set of ITP patients (*n* = 21) were used to monitor the changes of MP levels before and after TPO-RA treatment. Among them, 12 were responders and 9 were non-responders. The platelet counts of responders were significantly increased after TPO-RA treatment (*p* < 0.001), while the platelet counts of non-responders remained similar before and after the treatment (*p* > 0.05) (Table 1).

### 3.2. Plasma MP Levels in ITP, TP Patients and Healthy Controls

As shown in Figure 2A, both MKMP and PMP levels in ITP patients were significantly lower than those in healthy controls (HC) (*p* values < 0.0001), but higher than those in TP patients (*p* < 0.0002 and *p* < 0.002, respectively). To exclude the effect of platelet counts on MP levels, PMP levels were normalized to PMP/platelet count (PLT) ratios. The values of the PMP/PLT ratio in ITP patients were significantly higher than those in HC (*p* < 0.0001) and TP patients (*p* < 0.0002) (Figure 2B). Interestingly, PMP/PLT ratios appeared to have a bimodal distribution pattern in all three groups. It remained unclear if the individuals with greater PMP/PLT ratios had higher percentages of newly formed platelets that tended to be bigger in size and/or more sensitive to activating stimuli, thereby producing more PMPs.

ITP patients in the discovery set were divided into several subgroups according to disease course, presence or absence of platelet autoantibodies, and antibody subtypes (Appendix A). There were no significant differences observed among these subgroups.

### 3.3. Diagnostic Values of Plasma MPs for ITP

To evaluate the sensitivity and specificity of plasma MP levels for ITP diagnosis, ROC curves were constructed with the data of the discovery set (Figure 3).

AUC and cut-off values for PMPs, MKMPs, and PMP/PLT ratios to diagnose ITP and differentiate ITP from TP were listed in Table 2. Among them, PMP/PLT ratio had the highest diagnostic potential with 73.1% sensitivity and 77.3% specificity. In addition, MKMP levels at a threshold of 112.5 MPs/μL discriminated ITP from non-ITP with 74.4% sensitivity and 68.1% specificity (Table 2).

### 3.4. Elevated Plasma MKMP and PMP Levels in TPO-RA Responders

In the therapy set, plasma PMP and MKMP levels were increased significantly in ITP patients who responded to TPO-RA treatment (*p* < 0.05 and *p* < 0.005, respectively) (Figure 4A,B). In contrast, PMP and MKMP levels in non-responders remained similar before and after TPO-RA treatment (both *p* values > 0.05) (Figure 4C,D).

### 3.5. Correlation between Plasma MP Levels and ITP Parameters

No significant correlations were found between plasma PMP or MKMP levels and platelet counts in the discovery set (*r* = 0.101, *p* > 0.05; *r* = 0.199, *p* > 0.05, respectively) (Appendix A). Similarly, there were no significant correlations between plasma PMP or MKMP levels and percentage of reticulated platelets (rPLT%) (*r* = 0.092, *p* > 0.05; *r* = 0.129, *p* > 0.05, respectively) (Appendix A) or TPO levels (*r* = 0.144, *p* > 0.05; *r* = 0.046, *p* > 0.05, respectively) (Appendix A). However, there was a significant positive correlation between PMP levels and platelet counts in ITP patients who responded to TPO-RA treatment (Figure 5A). In contrast, no significant correlation was found between plasma PMP levels and platelet counts in TPO-RA non-responders (Figure 5C) or between MKMP levels and platelet counts in TPO-RA responders (Figure 5B) or non-responders (Figure 5D).

## 4. Discussion

ITP is a common bleeding disorder, whose diagnosis is primarily based on platelet counts and exclusion of morphology abnormalities in peripheral blood smear [25]. Accumulating evidence suggests that megakaryocyte abnormalities contribute to the pathogenesis of ITP. However, there are no peripheral blood-based tests to assess megakaryocyte dynamics for ITP diagnosis. To date, plasma microparticles have been identified as disease biomarkers [9,26,27,28]. We therefore hypothesized that circulating megakaryocyte- or platelet-derived MPs may be used as potential biomarkers for ITP diagnosis and/or prognosis.

Flow cytometry is a common method to measure and quantify cell-derived MPs of defined sizes. In this method, carefully selected instrumental setting and correct data analysis are critical [29]. In this population-based case-control study, we used flow cytometry with previously reported antigen markers to analyze plasma PMPs and MKMPs in ITP patients and healthy controls by ultracentrifugation and 2% Triton X-100 treatment [26] (Appendix A). To improve assay sensitivity, we compared MP event collection with Gallios cytometer under different discrimination settings. We found that more events were obtained under FL than the FS discrimination setting, consistent with previous reports [23,24].

Our results showed that plasma MKMP and PMP levels were decreased in ITP and TP patients compared to those in healthy controls and that the levels in ITP patients were higher than those in TP patients. We also found that the PMP/PLT ratio and MKMP level had high sensitivity and specificity values for ITP differential diagnosis. Moreover, plasma PMP levels positively correlated with platelet counts in ITP patients who responded to TPO-RA treatment. These results are consistent with the role of megakaryocyte dysfunction in the pathogenesis of ITP.

Previously, several studies reported increased plasma PMP levels in ITP patients compared to those in healthy controls [11,12,13,30]. There are several possibilities that may account for the apparent difference. Firstly, different methods for MP detection, particularly gating strategy and instrumental settings, may result in data variations [20,29,30]. Secondly, MPs may be defined differently in different studies. In our study, PMPs were defined as size < 1 μm Annexin V+CD41+CD62P+ MPs [6]. In other studies, PMPs were defined as size < 2 μm and CD61+ [12] or < 0.5 μm and CD42+ MPs [30], mostly without phosphatidylserine detection [11,12,13,30]. Thirdly, the data were collected from different patient populations, for example, pediatric ITP patients versus adult ITP patients in our study [12]. Further studies are important to verify our findings using standardized methods in similar patient populations.

To exclude the effect of platelet counts on PMP quantification, we included non-ITP patients (TP group) as another control and normalized PMP levels with platelet counts, i.e., PMP/PLT ratio. We found that MKMP and PMP/PLT ratios were significantly higher in ITP patients than those in TP patients. This may reflect that platelets are larger and more responsive to activating stimuli in ITP patients than in some TP patients (e.g., myelodysplastic syndromes and acute myeloid leukemia) [31]. Moreover, chemotherapy in some TP individuals (e.g., leukemia) may result in megakaryocyte defects that inhibit MKMP release. The higher PMP levels and PMP/PLT ratios may also explain the observation that ITP patients are less likely to experience severe bleeding than TP patients despite similar thrombocytopenia [11,12,32].

Circulating MPs are recognized biomarkers for disease diagnosis and assessment [5]. Our results showed that the PMP/PLT ratio had the highest AUC value of 0.808 for ITP diagnosis at a cut-off value of 0.165 × 10^−3^ with 73.1% sensitivity and 77.3% specificity. Additionally, plasma MKMP levels can differentiate ITP from non-ITP at a cut-off value of 112.5 MPs/μL. These findings indicated that plasma MKMP and PMP levels can serve as potential biomarkers for ITP differential diagnosis.

TPO-RAs are used as second-line therapies to promote megakaryocyte proliferation and maturation in ITP patients. Changes in plasma MKMP levels may reflect megakaryocyte status in the bone marrow, thereby facilitating the assessment of the response to TPO-RA treatment. We found that plasma MKMP and PMP levels in the responders increased significantly after TPO-RA treatment and that plasma PMP levels positively correlated with platelet counts during the course. In contrast, plasma MKMP levels in TPO-RA non-responders remained unchanged. This is consistent with the underlying mechanisms of TPO-RAs in promoting megakaryocyte proliferation and platelet production, thereby increasing circulating platelet counts [33,34]. Thus, monitoring MKMP levels may help to evaluate megakaryocyte status and identify responders to TPO-RA treatment, thereby avoiding unnecessary bone marrow aspiration procedures.

This study has several limitations. Firstly, we used Annexin V staining to distinguish MPs from other small particles, as recommended by ISEV. Some circulating MPs may not have surface phosphatidylserine and are susceptible to calcium ions in the buffer, which may increase data variations [22,35]. Secondly, the mechanistic link between the reduced plasma MKMP levels and megakaryocyte conditions in ITP patients remains to be determined. Thirdly, our study focused on plasma MKMP and PMP levels. We did not evaluate the effect of the altered MKMP and PMP levels on the overall thrombotic state in the ITP patients. Finally, our study cohorts are small. Further studies with larger cohorts and more therapeutic groups are needed to verify our findings regarding the diagnostic value of plasma MKMPs and PMPs in ITP.

## 5. Conclusions

Plasma MKMP and PMP levels were decreased in ITP patients. Plasma MKMPs and PMPs may serve as potential biomarkers for ITP diagnosis and prediction of TPO-RA treatment response. Our findings warrant further studies to understand the role of MPs in the pathogenesis of ITP.

## Figures and Tables

**Figure 1 jcm-11-06776-f001:**
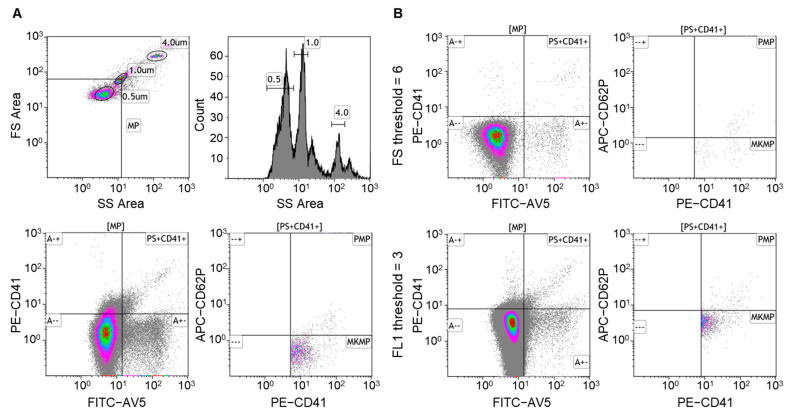
Flow cytometry analysis of plasma MPs: (**A**) Calibrated latex beads (0.5 and 1.0 μm) were used to determine the MP gate (particle size < 1.0 μm). Four μm counting beads were shown in the upper right corner of the FSC/SSC view. Annexin V^+^CD41^+^ events were analyzed in the MP gate. The numbers of PMPs (CD41^+^CD62P^+^) and MKMPs (CD41^+^CD62P^−^) were determined based on the counting beads; (**B**) MPs in PFP samples were analyzed by Gallios Flow Cytometer under two discrimination settings: forward light scatter (FS) threshold = 6 (top panels) and fluorescence parameter 1 (FL1) threshold = 3 (bottom panels). More MP events were detected using FL1 threshold than FS threshold. PMPs, platelet-derived microparticles; MKMPs, megakaryocyte-derived microparticles.

**Figure 2 jcm-11-06776-f002:**
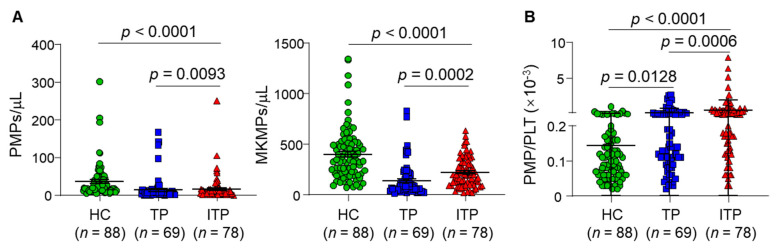
Plasma PMP and MKMP levels and PMP/PLT ratios in ITP patients: (**A**) Plasma profiles of MKMPs and PMPs in healthy controls (HC) (*n* = 88) and TP (*n* = 69) and ITP (*n* = 78) patients; (**B**) PMP levels were normalized to platelet counts as PMP/PLT ratios in HC (*n* = 88) and TP (*n* = 69) and ITP (*n* = 78) patients. Data are presented in the scatter plot view (Mean ± SEM).

**Figure 3 jcm-11-06776-f003:**
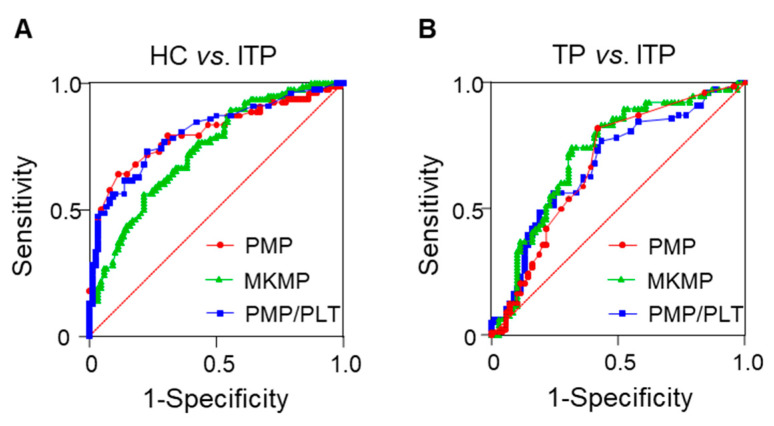
ROC analysis of plasma PMP and MKMP levels and PMP/PLT ratios for ITP diagnosis. ROC analysis was performed for PMP and MKMP levels and PMP/platelet count (PLT) ratios to evaluate their potential to distinguish ITP from healthy controls (**A**) and non-ITP patients (**B**). The AUC values were 0.805, 0.735 and 0.808 in (**A**) and 0.682, 0.723 and 0.681 in (**B**), respectively.

**Figure 4 jcm-11-06776-f004:**
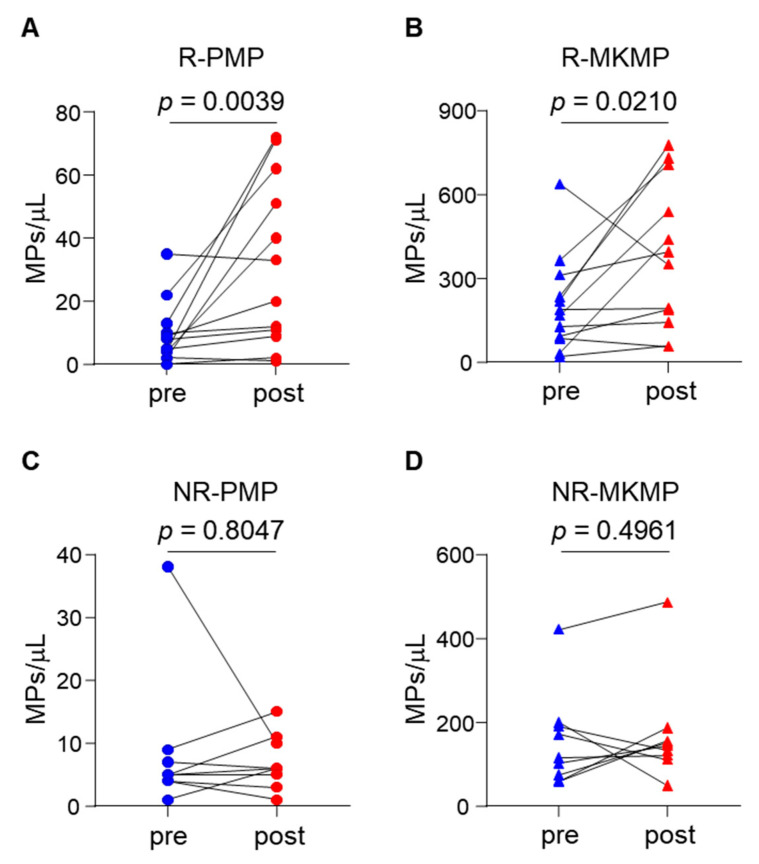
Plasma MPs levels before and after TPO-RA treatment. Plasma PMP and MKMP levels were measured before and after TPO-RA treatment in responders (**A**,**B**) and non-responders (**C**,**D**). Wilcoxon matched pairs *t*-test was performed for the comparisons between pre- and post-treatment groups.

**Figure 5 jcm-11-06776-f005:**
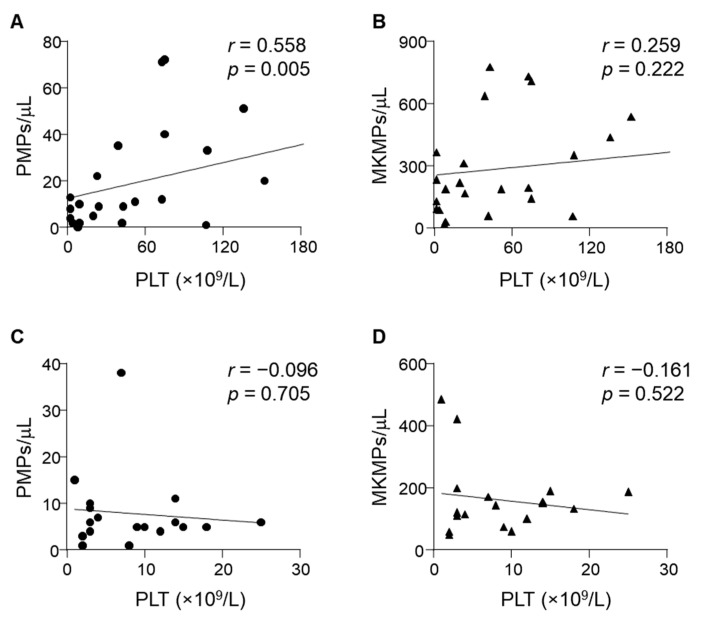
Correlation analysis between plasma MP levels and platelet counts. The Spearman correlation analysis was performed between plasma MP levels and platelet counts in responders ((**A**,**B**), *n* = 12) and non-responders ((**C**,**D**), *n* = 9) during the course of treatment. A positive correlation was found between PMPs and PLT counts in the responders ((**A**), *r* = 0.558, *p* = 0.005).

**Table 1 jcm-11-06776-t001:** Characteristics of the study participants.

Baseline Characteristics	Discovery Set			Therapy Set	
	HC (*n* = 88)	TP (*n* = 69)	ITP (*n* = 78)	Non-Responder (*n* = 9)	Responder (*n* = 12)
Age, yrs.	43 [37–50]	44 [33–58]	49 [34–58]	61 [31–68]	49 [27–59]
Male, *n* (%)	52 (59.1)	35 (50.7)	38 (48.7)	5 (55.6)	5 (41.7)
Disease course, *n* (%)					
Newly diagnosed	NA	NA	36 (46.2)	1 (11.1)	6 (50)
Persistent	NA	NA	17 (21.8)	5 (55.6)	1 (8.3)
Chronic	NA	NA	25 (32.1)	3 (33.3)	5 (41.7)
FCIA, *n* (%)					
Negative	NA	NA	53 (67.9)	5 (55.6)	10 (83.3)
GPIb	NA	NA	4 (5.1)	1 (11.1)	0 (0)
GPIIb	NA	NA	3 (3.8)	0 (0)	1 (8.3)
GPIIIa	NA	NA	7 (9.0)	1 (11.1)	0 (0)
Multiple Abs	NA	NA	11 (14.1)	2 (22.2)	1 (8.3)
Platelet count (×10^9^/L)					
Pre-treatment	238 [204–288]	24 [17–37] *	28 [10–56] *	7 [3–11]	9 [2–22]
Post-treatment	NA	NA	NA	8 [3–16]	75 [57–129] ^§^
RP %	NA	NA	19 [9.4–28.7]	11 [7.9–37.6]	23 [13.3–31.6]
MPV (fL)	10.5 [9.9–11.1]	12 [11.1–12.7] *	12.5 [11.6–13.9] *	11.2 [10.4–11.7]	11.4 [11.1–13.5]
TPO (pg/mL)	NA	NA	110.2 [52.2–172.8]	92.8 [41.4–147.8]	187 [169.4–279.7]
WBC (×10^9^/L)	6.3 [5.1–7.0]	1.5 [0.5–3.1] *	8.6 [6.3–11.1] ^#^	8.0 [5.0–10.9]	9.1 [6.8–10.5]
Hemoglobin (g/L)	145 [134–154]	75 [59–93] *	125 [110–140] *	125 [113–138]	131 [84–139]
Treatment, *n* (%)					
Steroids	NA	NA	77 (98.7)	9 (100)	12 (100)
Immune suppressant	NA	NA	11 (14.1)	4 (44.4)	2 (16.7)
Splenectomy	NA	NA	1 (1.3)	0 (0)	1 (8.3)
Others	NA	NA	70 (89.7)	9 (100)	12 (100)

Quantitative variables are expressed as median [interquartile range]. Categorical variables are presented as numbers (percentages). NA, not available. HC, healthy control; TP, non-immune thrombocytopenia; ITP, Immune thrombocytopenia; GP, glycoprotein; Abs, antibodies; FCIA, Flow cytometric immunobead assay; RP, Reticulated platelet; MPV, mean platelet volume; TPO, Thrombopoietin; WBC, white blood cell. Steroids referred to dexamethasone, prednisone and methylprednisolone; Immune suppressants referred to cyclosporine A (CSA), Azathioprine (Aza), Rapamycin (RPM) and Mycophenolate Mofetil (MMF); Others referred to intravenous immunoglobulin (IVIg), TPO, thrombopoietin receptor agonist (TPO-RA) and Rituximab. * *p* < 0.0001 compared to HC, ^#^
*p* = 0.0003 compared to HC, ^§^
*p* = 0.0005 compared to pre-treatment.

**Table 2 jcm-11-06776-t002:** ROC analysis of plasma MP levels for ITP differential diagnosis.

Parameters	HC vs. ITP			TP vs. ITP		
	PMP	MKMP	PMP/PLT	PMP	MKMP	PMP/PLT
AUC	0.805	0.735	0.808	0.682	0.723	0.681
95% CI	0.736–0.874	0.661–0.810	0.741–0.875	0.593–0.770	0.639–0.808	0.593–0.768
Cut-off values	<9.500 (MPs/μL)	<209.0 (MPs/μL)	>0.165 (×10^−3^)	>3.500 (MPs/μL)	>112.5 (MPs/μL)	>0.145 (×10^−3^)
Sensitivity (%)	64.1	56.4	73.1	82.1	74.4	76.9
Specificity (%)	88.6	78.4	77.3	58.0	68.1	56.5
*p*-value	<0.0001	<0.0001	<0.0001	0.0001	<0.0001	0.0002

AUC: area under the curve; CI: confidence interval. TP, non-immune thrombocytopenia; ITP, Immune thrombocytopenia.

## Data Availability

All the data support the findings of this study are presented in the manuscript and the Appendix A of this article.

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
