# Peer review of "Megakaryocyte- and Platelet-Derived Microparticles as Novel Diagnostic and Prognostic Biomarkers for Immune Thrombocytopenia"

_jcm, 2022, doi:10.3390/jcm11226776_

Round 1
Reviewer 1 Report
In this manuscript, Wang et al. has investigated the potential diagnostic and prognostic role of megakaryocyte- (MKMP) and platelet-derived microparticles (PMP) in immune thrombocytopenia (ITP).
However, some issues should be addressed.
First, comparison of clinical characteristics between groups should be performed (results should be added to Table 1).
Second, in Figura 2B, for all three groups, there are some patients/HC who showed higher ratios. Are there any explanation for this? It looks like it is not a random phenomenon, while there is a bimodal distribution.
Please include in the subgroup analysis also disease severity.
Other clinical characteristics should be included in the analysis not only platelet counts and TPO levels, such lymphocytes count.
Correlations with Pearson analysis and multivariate analysis should be performed.
Reviewer 2 Report
Wen Wang et al. to evaluate the role of altered cell-derived microparticles (MPs) of megakaryocyte and platelets in ITP; they examined plasma MKMP and PMP levels in patients with ITP and evaluated their potential diagnostic values.
The authors must clarify a series of aspects of this study, in particular some that concern the methodology used.
1. The authors must present a hypothesis regarding MP results in patients with ITP. What was the working hypothesis? Considering that patients with ITP have a consumption of platelets, the hypothesis could be that PM would be increased.
2. ITP is also considered a prothrombotic disease, in this sense, the PM could also have an interesting role. However the authors have not addressed this issue at all.
3. What the authors mean with “discovery set”. New diagnosed? Please clarify.
4. Was there a cut-off for the number of platelets to include the patients in the study?
5. Did patients with persistent or chronic forms of ITP have a previous treatment? There was a wash-up before included in the study? or had not been treated before.
6. How long the drugs were administered to consider that a patient was not a responder. These data should be described in the methodology.
7. In the non-immune thrombocytopenia group, the authors have to describe what underlying diagnosis the patients had.
8. Why data from control group and non-immunological thrombocytopenia are not shown in the table 1?
9. Plasma MP levels in ITP, TP patients and healthy controls. The authors show the interpretation of statistic analysis, but they do not show the values (representation in the figures is not enough). Values should be showed in numbers.
10. Diagnostic values of plasma MPs for ITP. The only curve that has value is the comparison between TP and ITP (graphic 3.3.B). Since they are talking about diagnostic value, MP values in patients with normal platelets have no value in this type of curves.
11. Due to the low number of patients evaluated as responders vs non-responders, these data should be interpreted with greater care and need confirmation. This should be specially addressed in the discussion.
12. It would be interesting to include a longitudinal follow-up of some of the patients, showing the MPs data in correlation with the number of platelets and other parameters and showing the treatment.
13. The authors mentioned about the use of this test in the diagnosis of ITP. In this aspect, they should suggest when to carry out the investigations and the problems in interpretation if the test is done when the patients have improved their number of platelets.
Round 2
Reviewer 1 Report
The Authors have addressed all comments.